# Dietary Intake in Early Pregnancy and Glycemia in Late Pregnancy among Women with Obesity

**DOI:** 10.3390/nu14010105

**Published:** 2021-12-27

**Authors:** Kirsti Krohn Garnæs, Trude Elvebakk, Øyvind Salvesen, Signe Nilssen Stafne, Siv Mørkved, Kjell Åsmund Salvesen, Trine Moholdt

**Affiliations:** 1Department of Obstetrics and Gynaecology, St. Olavs Hospital, Trondheim University Hospital, 7006 Trondheim, Norway; kirsti.k.garnas@ntnu.no (K.K.G.); pepe.salvesen@ntnu.no (K.Å.S.); 2Department of Public Health and Nursing, Faculty of Medicine and Health Sciences, Norwegian University of Science and Technology, NTNU, 7089 Trondheim, Norway; signe.n.stafne@ntnu.no (S.N.S.); siv.morkved@ntnu.no (S.M.); 3Children’s Clinic, St. Olavs Hospital, Trondheim University Hospital, 7006 Trondheim, Norway; trude.elvebakk@stolav.no; 4Faculty of Medicine and Health Sciences, Faculty Administration, Norwegian University of Science and Technology, NTNU, 7089 Trondheim, Norway; oyvind.salvesen@ntnu.no; 5Department of Clinical Service, St. Olavs Hospital, Trondheim University Hospital, 7006 Trondheim, Norway; 6Department of Clinical and Molecular Medicine, Norwegian University of Science and Technology, NTNU, 7089 Trondheim, Norway; 7Department of Circulation and Medical Imaging, Norwegian University of Science and Technology, NTNU, 7089 Trondheim, Norway

**Keywords:** maternal health, early pregnancy diet, maternal glucose, gestational diabetes mellitus, GDM

## Abstract

Women with obesity have increased risk for hyperglycemia during pregnancy, with negative health consequences for mother and child. We aimed to investigate adherence to nutritional recommendations in early pregnancy and to examine associations between early pregnancy dietary intake and late pregnancy glycemia among women with obesity. We included 120 women with pre-pregnancy body mass index (BMI) ≥30 kg/m^2^ who participated in one of two randomized controlled trials. The participants completed a food frequency questionnaire in early pregnancy (gestational weeks 12–22). Fasting and 120 min glucose tolerance after ingestion of 75 g glucose were assessed in late pregnancy (gestational weeks 32–37). About 90% of the participants reported early pregnancy diary intake within the recommendations. Average intakes of vitamin D, iron, and folate were below recommended levels. High intakes of dairy products and protein in early pregnancy were associated with lower fasting glucose in late pregnancy, whereas high intake of bread was associated with lower 120 min glucose. There were no clear associations between single dietary variables and gestational diabetes mellitus (GDM) diagnosis in late pregnancy. In conclusion, some early pregnancy dietary variables were associated with late pregnancy glycemia. Potential causality of these findings should be investigated in future studies.

## 1. Introduction

Maternal obesity is associated with hyperglycemia, which in turn is associated with increased risk of adverse pregnancy outcomes for mother and child. Currently, about one third of women have pre-pregnancy body mass index (BMI) ≥30 kg/m^2^, and the proportion of pregnant women with obesity is expected to increase [1]. Obesity and impaired maternal glucose metabolism are strongly associated with gestational diabetes mellitus (GDM) [2], macrosomia, and infant insulin resistance [3,4,5,6]. GDM is defined as glucose intolerance less than overt diabetes, with onset or first recognition during pregnancy [7]. GDM substantially increases the mother’s risk of progression to type 2 diabetes [8]. Additionally, infants born to women with reduced insulin sensitivity have increased risk of obesity and insulin resistance during childhood [6,9].

Lifestyle modification, including diet and exercise, has been proposed as the first-line strategy to prevent GDM, but the evidence for its effectiveness remains inconclusive [10,11,12]. Observational studies suggest that diet before and during pregnancy influence the risk of GDM [13,14,15,16], but the results are conflicting [17]. A systematic review of observational studies indicated that higher intakes of fat, heme iron, cholesterol, eggs, and red and processed meat were associated with increased GDM risk, whereas dietary patterns rich in fruit, vegetables, whole grains, and fish, and low in red and processed meat, refined grains, and high-fat dairy were beneficial [18]. The Mediterranean diet is associated with not only reduced GDM risk but also lower blood glucose levels in women with established GDM [19].

We have previously analyzed dietary data at early and late pregnancy from 702 participants who participated in the Norwegian Training in Pregnancy (TRIP) trial (*n =* 855), a randomized controlled trial (RCT) on exercise training in pregnancy [20,21]. Participants in all BMI categories were included in the dietary analyses, with 81% of the participants having a pre-pregnancy BMI <25 kg/m^2^. We found no dietary differences between participants who developed GDM (*n* = 40) and those who did not (*n* = 662); nor were there any associations between dietary variables in early or late pregnancy and blood glucose variables [21]. Previous research has indicated that diet before pregnancy is particularly important for glycemia in pregnancy [22], and women who plan a pregnancy should ideally implement a healthy diet. However, not all pregnancies are planned or conceived in proper time to implement healthy habits. It is important to identify early pregnancy nutritional factors associated with increased risk of GDM to establish effective strategies for prevention of adverse maternal and neonatal outcomes [23].

In the present study, we investigated the dietary intake of participants with obesity included in the TRIP trial, combined with data from participants with obesity in the Exercise Training in Pregnancy (ETIP) trial, which was an RCT on exercise training for women with pre-pregnancy BMI ≥28 kg/m^2^ [24]. The aim of the current study was to assess associations between early pregnancy dietary intake and fasting glucose, glucose tolerance, and GDM diagnosis in late pregnancy in women with pre-pregnancy obesity. We used the Nordic Nutrition Recommendations to describe the participants’ intake of nutrients [25].

## 2. Materials and Methods

### 2.1. Study Design and Participants

We included participants from the TRIP and ETIP trials with pre-pregnancy obesity (BMI ≥ 30 kg/m^2^) and dietary data from early pregnancy. The TRIP and the ETIP trials were RCTs that determined the effects of supervised exercise during pregnancy on maternal and fetal health. For the current study, we included participants independent of their group allocation in TRIP and ETIP. Previous publications have described the study design and primary findings of TRIP [20] and ETIP [24]. Pregnant women were recruited from St. Olavs hospital, Trondheim University Hospital, and Stavanger University Hospital between 2007 and 2009 (TRIP) and from St. Olavs hospital between 2010 and 2015 (ETIP). Both hospitals are run by the public healthcare trusts.

During 2007–2009, an invitation to participate in an exercise RCT (TRIP) to prevent GDM, lumbopelvic pain, and other pregnancy-related diseases was sent, along with an invitation to a routine ultrasound scan at gestational weeks 17–19. The ultrasound scan, which is free of charge and offered to all pregnant women in Norway, has 98% attendance [26]. Women who were willing to participate consented at the ultrasound appointment, and eligible women were enrolled the following week. The eligibility criteria in 2007–2009 included maternal age ≥18 years, Caucasian ethnicity, a singleton pregnancy, and living ≤30 min drive from the two hospitals. The primary recruitment strategy for the ETIP trial in 2010–2015 was identical to that for TRIP. Women with a self-reported pre-pregnancy BMI ≥28 kg/m^2^ and who were physically inactive (defined as habitual exercise training less than twice weekly before the current pregnancy) were invited to participate in an exercise RCT to limit gestational weight gain. Additional eligibility criteria in 2010–2015 included maternal age ≥18 years and a singleton pregnancy. The invitation to participate was also given to women attending fetal diagnostics, which is usually undertaken in gestational weeks 11–14 in Norway.

Exclusion criteria in both recruitment periods were high-risk pregnancies and/or diseases that could interfere with participation, including a history of giving birth before 34 weeks gestation, preeclampsia, severe fetal growth restriction, asthma, heart disease, renal disease, known substance abuse, placenta previa, blood pressure >140/90 mmHg on two or more measurements before gestational week 20, and/or identified high risk of preterm labor.

The participants were randomly allocated to either a supervised exercise program following the American College of Obstetricians and Gynecologists recommendations for physical activity during pregnancy [27] or a control group. All participants received standard prenatal care, which in Norway is universal and offered through the national health services. All participants received a brochure with guidelines about healthy nutrition in pregnancy as recommended by the Norwegian Directorate of Health [25]. The participants were not compensated financially.

The two trials from which the current participants were included are registered in ClinicalTrials.gov (TRIP; NCT00476567, ETIP; NCT01243554) and approved by the Regional Committee for Medical and Health Research Ethics (TRIP; REK 4.2007.81, ETIP; REK-midt 2010/1522).

### 2.2. Outcome Measures

#### 2.2.1. Dietary Intake

Dietary data were collected at enrolment (early pregnancy, gestational weeks 12–22) with a food frequency questionnaire (FFQ).

The FFQ is quantitative, self-administered, optical mark readable, and validated against the Norwegian adult population [28,29]. We used the 1997 version of the FFQ in TRIP and the 2007 version in ERIP. The wording in the FFQ asks respondents to report “average” of their dietary habits and to keep the last year in mind when responding. The participants were instructed to record their dietary intake the previous four weeks instead. The FFQ was handed out and filled in at the hospitals while the participants waited for blood glucose analysis.

The FFQ provides data on daily intakes of foods and beverages and includes questions on 180 food items, including nutritional supplements. Depending on the food item, the participants selected alternatives for frequency of intake of each food per day, per week, or per month. Portion sizes were specified in household units, which were later computed into edible amounts of food based on standardized scales. Related food items were categorized into 15 main groups of foods and beverages and reported as daily intake in grams (g/day). Estimations of total daily energy intake and nutrients were computed using a food database and software based on the official Norwegian Food Composition Table [30] at the Institute of Basic Medical Sciences, University of Oslo. Factors for energy distribution (E%) calculations of macronutrients were as follows: 4 kcal/g for protein and carbohydrates (minus fiber), 9 kcal/g for fat, 7 kcal/g for alcohol, and 2 kcal/g for fiber.

We used the 2012 Nordic Nutrition Recommendations (NNR) to describe energy distribution, fiber intake, and intake of micronutrients [25].

#### 2.2.2. Blood Sampling and Biochemistry

Early pregnancy assessments were undertaken at baseline in gestational weeks 12–22, and late pregnancy assessments were undertaken in gestational weeks 32–37. At both timepoints, participants underwent a 120 min oral glucose tolerance test (OGTT) with a 75 g glucose load. Blood samples were analyzed according to standard laboratory procedures, as described previously [24]. In the current analyses, we used the 2013 WHO and the International Association of Diabetes and Pregnancy Study Groups (IADPSG) diagnostic criteria for classification of GDM as follows: fasting plasma glucose ≥5.1 mmol/L and/or 2 h plasma glucose ≥8.5 mmol/L [7,31].

#### 2.2.3. Other Measurements

Data on pre-pregnancy weight, height, and level of exercise before pregnancy were self-reported in questionnaires. Body weight was measured by study personnel at baseline and in late pregnancy to the nearest 0.1 kg. All women were weighed while fasting, without shoes, and with light clothing. Height was self-reported in 2007–2009 and measured at early pregnancy in 2010–2015 by a wall-mounted stadiometer. BMI was calculated as weight in kilograms divided by the square of height in meters.

### 2.3. Ethical Considerations

The two trials from which the current participants were recruited were conducted in accordance with ethical standards of research and the Helsinki Declaration. The research questions in the current paper are within the research scope of which the participants gave informed, written consent.

### 2.4. Statistical Methods

Participant characteristics and demographic variables are reported as mean and standard deviation (SD) or number (*n*) and percentage (%) as appropriate. Due to skewed dietary data, dietary variables are reported as group means and medians with 25th and 75th percentiles.

The associations between early pregnancy dietary intake and late pregnancy fasting glucose and 120 min glucose were analyzed using log-linear regression, whereas the associations between early pregnancy dietary intake and GDM diagnosis were analyzed using logistic regression. The dietary variables were analyzed one at a time, each time adjusting for gestational age, early pregnancy BMI, and maternal age. A possible interaction between group allocation and diet on late pregnancy glucose levels was examined by including group allocation (categorial variable) into the log-linear model, and no statistically significant interactions were found. Results are reported as coefficient (*β*) for log-linear models and odds ratios for logistic models with 95% confidence intervals (CIs). The coefficient represents the effect (on log scale) on fasting glucose/120 min glucose/GDM per one unit increase in the respective dietary variable. Numeric results are presented with coefficient (*β*) and 95% CI.

Due to the high number of comparisons and risk for type I error, we considered *p*-values <0.01 as statistically significant. IBM SPSS Statistics for Windows, version 25 (IBM Corp., Armonk, NY, USA) and R version 3.6.3, with package lme4 (R Foundation for Statistical Computing, Vienna, Austria) were used for statistical analysis.

## 3. Results

### 3.1. Study Population

From the TRIP trial (2007–2009), 37 women fulfilled the inclusion criteria of pre-pregnancy BMI ≥30.0 kg/m^2^ and dietary data from early pregnancy. In ETIP (2010–2015), 83 women fulfilled the criteria. Thus, we included 120 participants in the analyses at baseline and had late pregnancy data from 95 participants (Figure 1).

Table 1 shows the characteristics of the participants, early and late pregnancy blood glucose levels, and late pregnancy prevalence of GDM. Of our study population, 48% were allocated to the exercise intervention group in the respective trials. Sixteen participants developed GDM.

### 3.2. Associations between Early Pregnancy Dietary Intake and Late Pregnancy Blood Glucose

Table 2 shows the associations between early pregnancy dietary intake and blood glucose and GDM diagnosis in late pregnancy. Dairy products, protein (g/day), cholesterol, calcium, and magnesium were inversely associated with fasting glucose in late pregnancy, while bread was inversely associated with 120 min glucose. No significant associations were found between dietary variables and GDM diagnosis. There were tendencies of inverse associations between fish/shellfish, eggs, and fiber and fasting glucose; between EI and riboflavin and 120 min glucose; and between eggs, alcohol (g/day), and alcohol E% and GDM. These associations had *p*-values from 0.013 to 0.049.

### 3.3. Dietary Intake in Early Pregnancy

Table 3 shows the daily intakes of foods, beverages, and nutrients in early pregnancy according to the Nordic Nutrition Recommendations (2012) [25] for energy distribution and daily intake of micronutrients. The median EI in early pregnancy was 2923 kcal. More than half of the participants had an estimated energy distribution within recommended ranges, with 98 (82%), 104 (87%), 81 (68%), 99 (86%), and 71 (59%) participants meeting the recommended ranges of E% for fat, protein, carbohydrates, added sugar, and fiber, respectively. Fifty-eight (50%) participants reported some alcohol consumption in early pregnancy. Fifty-nine participants reported no alcohol consumption. Hence, the median value for alcohol intake in early pregnancy was 0 (Table 3). Of those who reported alcohol consumption, 16 participants (28%) reported an intake of pure alcohol 0.10–0.49 g/day, 33 (57%) reported an intake of 0.5–6.0 g/day, 4 (7%) reported an intake of 6.1–11.9 g/day, and 5 (9%) reported an intake of ≥12 g/day. Converted into alcohol units (1 unit = 12 g pure alcohol), the average daily intake was 0.27 unit (range 0.01–1.63 units).

More than 75% of the participants consumed the recommended intake of vitamin A, vitamin C, riboflavin, and magnesium. Seventy-five participant (62%) reported consuming the recommended intake of calcium, 20 participants (27%) met the recommendation for intake of vitamin D, 25 (21%) for iron, and 29 (25%) for folate. Cholesterol intake was in line with the intake in Nordic countries [25].

## 4. Discussion

### 4.1. Short Summary of Findings

We assessed associations between early pregnancy dietary intake and glycemia in late pregnancy in women with pre-pregnancy obesity. We found inverse associations between dairy products, protein (g/day), cholesterol, calcium, and magnesium and fasting glucose and between bread and 120 min glucose. No significant associations were found between dietary variables and GDM. Most participants had energy distribution and fiber intake within the recommended ranges [25] and met the dietary recommendations for most micronutrients, but the intakes of vitamin D, iron, and folate were lower than recommended. Half of the population reported alcohol consumption in early pregnancy, but the median reported intake of those reporting alcohol consumption was low.

### 4.2. Strengths

The FFQ used in the current study has been validated in the Norwegian population. Further, we analyzed a wide range of dietary variables, from eligible amounts of consumed foods to actual amounts of macronutrients, estimates of micronutrients, and distribution of energy from each macronutrient. FFQs provide information about episodically consumed foods, which made it a suitable instrument for measuring the habitual intake the previous month during the current pregnancy. Our study population was merged from two trials with identical protocols regarding intervention, data collection, and follow-up.

### 4.3. Limitations

Dietary intakes were self-reported, and social desirability bias and recall bias may have affected dietary reports. Misreporting of dietary intakes is a well-known pitfall of dietary research, and FFQs deviate more from measured energy biomarkers than other dietary assessment methods [32]. Dietary assessment was not the primary aim of the trials from which the current participants were recruited; thus, the method of choice had to provide low respondent burden, be of low cost, and be easy to administer. The FFQ used in our study has been validated in the general population but not in pregnant women. Dietary characteristics and/or changes related to pregnancy symptoms are thus perhaps not sufficiently captured; however, it is possible to assume that our population was quite healthy as the participants volunteered for an exercise trial. We lacked data on fatty acid composition and subgroups of foods (e.g., red meat versus poultry). Half of the current population was randomized and exposed to an exercise intervention between the early pregnancy dietary assessment and the OGTT in late pregnancy. Group allocation in the two trials did not statistically affect the associations between early pregnancy diet and late pregnancy blood glucose, but the exercise intervention may have modified the effect sizes of the respective associations. Furthermore, changes in dietary intake and physical activity throughout pregnancy may have affected glucose levels in late pregnancy. We did not include measurements of serum levels of vitamin D, serum ferritin, or serum folic acid; thus, the prevalence of insufficiencies or deficiencies of either micronutrient is unknown. To limit the risk of false-positive findings due to multiple comparisons, we only interpreted *p*-values ≤0.01 as statistically significant. We acknowledge that this could have led to a type 2 error for the associations with *p*-values between 0.01 and 0.05.

### 4.4. Generalizability

According to data from the Norwegian Medical Birth Registry [33], the participants in our study are representative of Norwegian pregnant women with obesity in terms of obesity grades, age, education, parity, and occupational activity/employment [24]. All participants volunteered for an exercise intervention during pregnancy and might have been more health conscious than the general population of pregnant women with obesity.

### 4.5. Comparison with Previous Studies

#### 4.5.1. Early Pregnancy Diet and Late Pregnancy Glycemic Control

In our participants, intake of dairy products and protein (g/day) was associated with lower fasting glucose in late pregnancy. There are indications in the literature that a high protein intake combined with carbohydrate restriction during pregnancy may improve glycemic control [34], but the evidence is limited thus far. Several dairy components, such as milkfat, vitamin D, calcium, magnesium, potassium, and whey proteins, may individually or collectively have a beneficial effect on fasting glucose [35].

We found inverse associations between the intake of magnesium and calcium and fasting glucose. Magnesium has several beneficial effects, but the effect of dietary magnesium on glucose metabolism during pregnancy has not been studied much. In one small diet-controlled, randomized trial of 1000 IU vitamin D and 1000 mg calcium supplementation versus 250 IU vitamin D and 500 mg calcium supplementation in women with GDM, the group that received the largest doses had significant reductions in several markers of glucose metabolism, including fasting glucose and insulin concentrations after six weeks [36]. The lower doses of these supplements also reduced fasting glucose but had no effect on insulin concentrations.

Dietary cholesterol was associated with lower fasting glucose in late pregnancy. This is contrary to the findings of a Chinese study, in which dietary cholesterol, particularly from eggs, was associated with higher fasting glucose in late pregnancy and greater GDM risk [37]. Studies from the US also reported that high egg and cholesterol intakes before and during pregnancy were associated with increased risk of GDM [38]. In contrast, we observed tendencies of lower fasting glucose (*p* = 0.049) and lower risk of GDM (*p* = 0.017) associated with the intake of eggs in early pregnancy. These conflicting findings may be due to variations in dietary patterns in the studied populations and the different times the data were collected. The US studies [38] were conducted in the 1990s and early 2000s, when a limited egg intake was recommended, so a high intake of eggs during this time period could potentially reflect low adherence to dietary recommendations. We found no significant associations between foods and nutrients in early pregnancy and GDM diagnosis in late pregnancy, which is supported by a prospective pregnancy and birth cohort study in the US [17] and our previous dietary analyses of the entire TRIP population [21]. In fact, Radesky et al. [17] suggested that nutritional status when entering pregnancy, as reflected by pre-pregnancy BMI, is probably more important than diet during pregnancy for GDM risk. This suggestion is supported by Zhang et al. [22], who showed that women with a healthy lifestyle (BMI <25 kg/m^2^, sufficient physical activity, healthy eating, and not smoking) during pre-pregnancy had 83% lower risk of GDM compared with women who did not meet these criteria for a healthy lifestyle.

#### 4.5.2. Early Pregnancy Dietary Intake

The average EI (mean 3248 kcal/d, median 2860 kcal/d) in the current study was higher than what was estimated in a previous dietary analysis of the entire TRIP population (median ≈ 2100 kcal/day), in which 81% of the participants had pre-pregnancy BMI <25 kg/m^2^ [21]. A meta-analysis of studies conducted in North America, Europe, and Japan reported a mean EI of 1842–2213 kcal/day during pregnancy [39], which was similar to the whole TRIP population [21]. The higher EI of the current population does not seem to be a result of intake of energy-dense and/or unhealthy foods, as supported by the majority of participants having an energy distribution and fiber intake within recommended ranges. Compared to the Norwegian female population aged 30–39 years [40], our participants consumed less cake, sugar, and sweets; less of the “various intakes” category, which includes energy-dense snacks; and more vegetables. As shown by a higher energy intake, the consumption of other food groups was higher, especially cereals and dairy products. In all, it appears that the participants made deliberate choices toward a healthy diet. A comparison with the national survey is suitable because food and energy variables were computed based on the Norwegian food database [30] in our study and the national survey and reported equally. The national survey used 24 h recalls, which is a more accurate method.

Fifty-eight (50%) participants reported alcohol consumption in early pregnancy; the corresponding prevalence in dietary analyses of the whole TRIP population was 35% [21]. This is surprising as alcohol consumption is usually terminated when a pregnancy is recognized [41,42]. In Norway, 4.1% reported continuing consumption of alcohol after pregnancy recognition, which is a lower prevalence than many other European countries [43,44]. It is suspected that pregnant women tend to underreport their alcohol consumption during pregnancy [44], but the low rate of alcohol use during pregnancy in Norway is supported by analysis of the biomarker phosphatidylethanol (PEth). In an unselected population of 4067 pregnant women, 1.4% had a positive PEth sample around gestational week 12, whereas 0.4% had a positive sample around week 24 [45]. Methodological bias could possibly explain the large proportion reporting alcohol consumption in the current study as the wording in the FFQ was not specific about current alcohol consumption after the current pregnancy was recognized, which could potentially cause some participants to report their usual alcohol use before pregnancy recognition. Another possibility, which ought to be investigated further due to public health concerns, is that our selected population deviated from the general population of Norwegian pregnant women regarding alcohol use during pregnancy.

Lower than recommended intakes of vitamin D, iron, and folate has been observed in other pregnant populations [21,46,47] and across BMI categories [46]. Despite the current analysis including dietary supplements, the median intake of vitamin D was lower than recommended. Not only was the intake lower than the recommended 10 µg/day [25] but more than half of the participants did not even meet the estimated average requirement of 7.5 µg/day [25]. As dietary intake of vitamin D during pregnancy correlates with serum levels at northern latitudes [48,49], insufficient intake is a risk factor for insufficient serum levels. In previous analyses of the entire TRIP population, 34% of the participants had insufficient serum levels and 7% had vitamin D deficiency in late pregnancy [50]. Regarding iron, less than half of the population met the recommended daily intake of 15 mg, but nearly 75% met the estimated average requirement of 10 mg/day [25]. Iron deficiency anemia is the most common micronutrient deficiency globally [51]. Surveys suggest that up to 32% of European women of reproductive age have iron deficiency, with 5% having iron deficiency anemia [52].

Folate is a difficult biomarker to measure exactly; however, there is consensus that supplements are essential to reach the recommended intake of 400 µg/day [25]. In Norwegian prenatal care, folate supplementation is mainly emphasized during preconception and the first trimester, so it is possible that a greater proportion of the current population used supplements and thus met the recommended intake earlier in pregnancy.

## 5. Conclusions

Higher intakes of dairy products, protein, cholesterol, calcium, and magnesium in early pregnancy were associated with lower fasting glucose in late pregnancy among women with pre-pregnancy obesity. Most of the participants reported early pregnancy dietary intake within the recommendations, except for lower than recommended intakes of vitamin D, iron, and folate. Causal relationships between early pregnancy dietary intake and hyperglycemia in late pregnancy need to be established, and future studies on maternal nutrition, including alcohol intake, need to investigate associations between the level of intake, nutritional status, and maternal–fetal outcomes.

## Figures and Tables

**Figure 1 nutrients-14-00105-f001:**
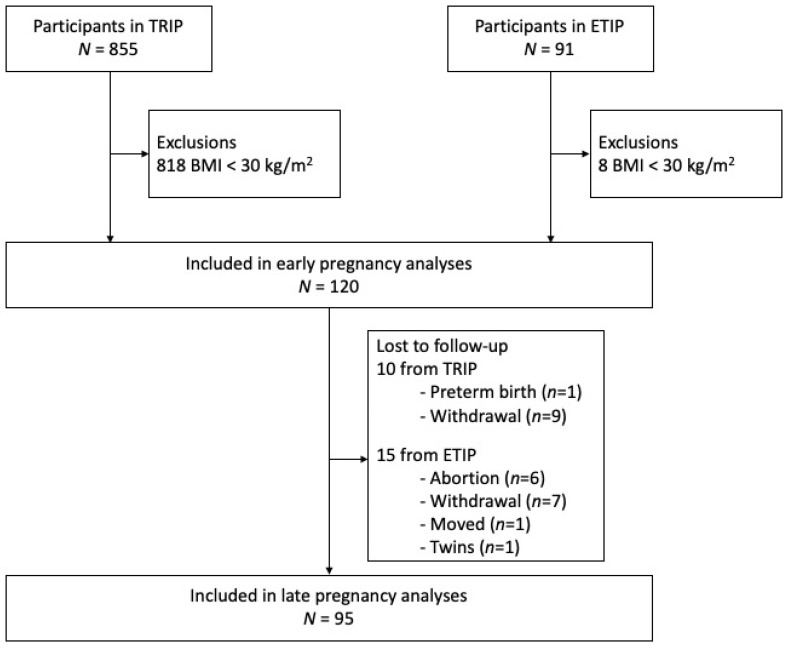
Flow chart of the participants included in the study.

**Table 1 nutrients-14-00105-t001:** Characteristics in early and late pregnancy of participants with pre-pregnancy obesity. Observed data are presented as mean and standard deviation (SD) or number of participants (*n*) and percentages (%).

Participants’ Characteristics	All Women Pre-Pregnancy *n* = 120	All Women Early Pregnancy * *n* = 120	All Women Late Pregnancy ** *n* = 95	Women with GDM Late Pregnancy ** *n* = 16
	Mean (SD)/*n* (%)	Mean (SD)/*n* (%)	Mean (SD)/*n* (%)	Mean (SD)/*n* (%)
Age, years		31.2 ± 4.3	-	33.3 ± 4.1
Education				
Primary/secondary school		3 (2.6)	-	
High school		35 (30.2)	-	
University ≤4 years		36 (31.0)	-	
University >4 years		42 (36.2)	-	
Currently employed		99 (83.2)	-	
Parity				
0		66 (55.0)	-	7 (43.8)
1		39 (32.5)	-	5 (31.3)
2		12 (10.0)	-	3 (18.8)
≥3		3 (2.5)	-	1 (6.3)
Current smoking		10 (8.5)	-	2 (12.5)
Gestational week	-	16.7 ± 3.7	34.9 ± 2.1	34.5 ± 1.8
Weight (kg)	95.2 ± 14.4	97.3 ± 12.5	104.0 ± 13.0	106.3 ± 16.3
Weight gain (kg) ***	-	-	7.1 ± 4.2	6.2 ± 6.0
Body mass index (BMI, kg/m^2^)	32.7 ± 4.5	34.5 ± 3.8	-	-
Obesity categories (pre-pregnancy), *n* (%)				-
Class 1 obesity, BMI 30.0–34.9 kg/m^2^	33 (35.5)	68 (57.1)	-	-
Class 2 obesity, BMI 35.0–39.9 kg/m^2^	42 (45.2)	36 (30.3)	-	-
Class 3 obesity, BMI ≥40.0 kg/m^2^	18 (19.4)	15 (12.6)	-	-
Glucose 0 (mmol/L)		4.7 ± 0.5	4.7 ± 0.5	5.4 ± 0.6
Glucose 120 min (mmol/L)		5.8 ± 1.2	6.2 ± 1.4	7.3 ± 1.8
Fasting insulin (pmol/L)		18.3 ± 9.1	28.0 ± 14.3	36.7 ± 21.5
Gestational diabetes mellitus, number (%)		-	16 (17.6)	-

* Gestational weeks 12–22. ** Gestational weeks 32–37. *** Late pregnancy weight minus early pregnancy weight (kg). Number of missing varies between 0 and 4 in early pregnancy and between 0 and 9 in late pregnancy.

**Table 2 nutrients-14-00105-t002:** Associations between early pregnancy dietary intake and fasting glucose, 120 min glucose, and GDM in late pregnancy. Results from univariate analysis are presented as estimated effect (beta), 95% CI, and *p*-value.

	Fasting Glucose			120 min Glucose			GDM		
Dietary Variable (Unit)	Estimated Effect (Beta)	95% CI	*p*-Value	Estimated Effect (Beta)	95% CI	*p*-Value	Estimated Effect (Beta)	95% CI	*p*-Value
Total energy intake (kcal)	−0.000011	−0.000025, 0.0000032	0.129	−0.00003	−0.000059, −0.000002	0.039	−0.0000546	−0.00011, 0.00001	0.072
Bread (g)	−0.00021	−0.00057, 0.00014	0.245	−0.00094	−0.00167, −0.00022	**0.010**	−0.000388	−0.0107, 0.00185	0.602
Cereals (g)	−0.00036	−0.000729, −0.000016	0.060	0.00010	−0.00068, 0.00089	0.798	0.001091	−0.00049, 0.00266	0.174
Cakes (g)	−0.00100	−0.00207, −0.0000774	0.068	−0.00019	−0.00229, 0.0020	0.859	−0.0021	−0.00649, 0.0022	0.333
Potatoes (g)	0.00026	−0.00038, 0.00091	0.422	0.00038	−0.0009, 0.00166	0.565	0.00068	−0.00319, 0.00182	0.593
Vegetables (g)	−0.000047	−0.000186, 0.0000933	0.513	−0.00012	−0.0004, 0.000163	0.401	−0.00018	−0.00076, 0.00039	0.532
Fruit and berries (g)	−0.00002	−0.000167, 0.000135	0.835	−0.00003	−0.00029, 0.00034	0.873	0.00019	−0.00041, 0.0008	0.531
Meat (g)	0.00023	−0.000215, 0.000683	0.308	−0.00034	−0.00125, 0.00057	0.466	0.00048	−0.0012, 0.0022	0.583
Fish and shellfish (g)	−0.00081	−0.00145, −0.000168	**0.013**	−0.0009	−0.0022, 0.00041	0.179	−0.00178	−0.0044, 0.00088	0.188
Eggs (g)	−0.00125	−0.0025, −0.0000025	0.049	−0.0010	−0.00364, 0.00149	0.412	−0.00569	−0.01039, −0.00099	0.017
Dairy products ^†^ (g)	−0.00009	−0.00014, −0.0000236	**0.006**	−0.00010	−0.00023, 0.00003	0.127	−0.000155	−0.00040, −0.00009	0.221
Cheese (g)	−0.00055	−0.00148, 0.000385	0.249	−0.0009	−0.00282, 0.00097	0.341	−0.00226	−0.00564, 0.0011	0.188
Butter, margarine, oil (g)	0.00074	−0.000396, 0.00187	0.201	−0.0020	−0.0043, 0.00028	0.085	0.0040	−0.00039, 0.0085	0.074
Sugar and sweets (g)	0.00005	−0.00072, 0.00082	0.898	−0.00098	−0.0025, 0.00056	0.213	0.000167	−0.00299, 0.0033	0.917
Beverages ^‡^ (g)	−0.000007	−0.000025, 0.000012	0.463	−0.00003	−0.00006, 0.000014	0.208	−0.000058	−0.00001, 0.000002	0.144
Various intakes ^§^ (g)	0.00019	−0.00028, 0.00067	0.463	0.00050	−0.00046, 0.00146	0.306	−0.00069	−0.0010, 0.00242	0.428
Protein (g)	−0.0011	−0.00188, −0.00039	**0.003**	−0.00158	−0.0031, −0.000010	0.051	−0.0017	−0.0048, 0.0013	0.263
Fat (g)	−0.00073	−0.0016, 0.00018	0.119	−0.00180	−0.00366, 0.00005	0.056	−0.0012	−0.0047, 0.0023	0.495
Fiber (g)	−0.00230	−0.0046, −0.0000475	0.045	−0.0025	−0.00743, 0.00240	0.315	−0.00058	−0.0103, 0.009	0.905
Carbohydrates (g)	−0.00029	−0.00058, 0.000001	0.051	−0.0005	−0.00107, 0.00014	0.137	0.0002	−0.00097, 0.0014	0.709
Sugar (g)	−0.00030	−0.0011, 0.0005	0.475	−0.0010	−0.0026, 0.0007	0.243	0.0014	−0.0018, 0.0047	0.395
Alcohol (g)	−0.00173	−0.0071, 0.00365	0.528	0.0044	−0.0064, 0.0151	0.430	−0.023	−0.045, 0.00083	0.042
Fat E%	0.00237	−0.0027, 0.0074	0.359	−0.00518	−0.0151, 0.0050	0.319	−0.0039	−0.023, 0.015	0.685
Protein E%	−0.00890	−0.0185, 0.00057	0.065	−0.0083	−0.0275, 0.0109	0.397	−0.0289	−0.066, 0.0084	0.128
Carbohydrates E%	0.00032	−0.00462, 0.00526	0.898	0.0055	−0.00438, 0.0153	0.276	0.01648	−0.0024, 0.0358	0.087
Sugar E%	−0.00070	−0.0176, 0.0059	0.754	−0.0079	−0.0051, 0.00103	0.453	0.0122	−0.0011, 0.0024	0.062
Alcohol E%	−0.00305	−0.0209, 0.0148	0.738	0.020	−0.0156, 0.0563	0.268	−0.080	−0.155, −0.0061	0.033
Cholesterol (mg)	−0.00029	−0.00049, −0.00008	**0.006**	−0.00036	−0.00080, 0.00007	0.099	−0.00051	−0.00132, −0.00028	0.905
Vitamin A (mcg)	−0.0000020	−0.000032, 0.000028	0.898	−0.000051	−0.0001, 0.000010	0.098	0.00009	−0.00003, 0.0002	0.140
Vitamin C (mg)	−0.0001	−0.00043, 0.00021	0.508	−0.00025	−0.0009, 0.0004	0.462	−0.000096	−0.00119, 0.00138	0.884
Vitamin D (mcg)	0.00069	−0.00469, 0.00608	0.801	−0.00026	−0.0135, 0.00819	0.629	0.00293	−0.0183, 0.02419	0.786
Riboflavin (mg)	−0.02317	−0.0492, 0.00286	0.081	−0.0622	−0.1161, −0.00827	0.023	−0.01119	−0.1135, 0.0911	0.830
Calcium (mg)	−0.000070	0.00011, −000030	**0.001**	−0.00008	−0.000165, 0.000010	0.083	−0.00013	−0.00029, −000037	0.128
Iron (mg)	0.000309	−0.00167, 0.00228	0.759	−0.00179	−0.00576, 0.00216	0.373	−0.0008	−0.009, 0.0074	0.436
Magnesium (mg)	−0.000239	−0.0004, −0.00007	**0.004**	−0.00033	−0.00069, 0.000018	0.063	−0.00028	−0.0009, 0.0004	0.436
Folate (mcg)	−0.000045	−0.00019, −0.000098	0.536	−0.000046	−0.00034, 0.00025	0.756	−0.000011	−0.00058, 0.00056	0.970

Statistics: Associations between early pregnancy dietary intake and late pregnancy fasting glucose and 120 min glucose were analyzed using log-linear regression, for GDM diagnosis, logistic regression. *p*-values <0.05 are indicated in bold font. The 120 min glucose was measured after an oral 75 g oral glucose tolerance test. Early pregnancy: gestational weeks 12–22; late pregnancy: gestational weeks 32–37. E%: percentage of energy from each macronutrient. Gestational diabetes mellitus according to the 2013 WHO and the International Association of Diabetes and Pregnancy Study Groups (IADPSG) diagnostic criteria for classification of GDM as follows: fasting plasma glucose ≥5.1 mmol/L and/or 120 min plasma glucose ≥8.5 mmol/L.

**Table 3 nutrients-14-00105-t003:** Daily intakes of main groups of foods, beverages, and nutrients, including supplements, in early pregnancy among women with pre-pregnancy obesity. Observed data presented as mean and median with 25th and 75th percentiles.

			Dietary Intake Early Pregnancy *n* = 120		
Edible Amounts Per Day		Mean	25th	Median	75th
Bread, g		193	149	188	236
Cereals, g		91	45	78	116
Cakes, g		23	8	19	28
Potatoes, g		55	29	55	74
Vegetables, g		266	146	243	373
Fruit and berries, g		280	184	245	347
Meat, g		120	83	110	146
Fish/shellfish, g		53	27	48	74
Eggs, g		24	11	22	34
Dairy products, ^‡^ g		516	267	459	687
Cheese, g		35	18	29	46
Butter, margarine, oils, g		30	15	27	41
Sugar and sweets, g		35	12	28	43
Beverages, ^§^ g		1780	1143	1596	2303
Various intakes, ^¶^ g		55	19	37	77
Energy and macronutrients per day					
Total energy intake, kcal	-	3293	2140	2923	4146
Total fat, g		86	66	79	103
Protein, g		95	73	91	112
Carbohydrates, g		267	217	248	312
Added sugar, g		45	26	38	54
	Recommendations *				
Fiber, g	25–35 g ^#^	29	22	28	36
Alcohol, g	Abstention **	1.58	0	0	1.15
Energy distribution					
Total fat E%	25–40 E% ^#^	35	31	34	38
Protein E%	10–20 E% ^#^	17	16	17	18
Carbohydrates E%	45–60 E% ^#^	48	44	48	51
Added sugar E%	<10 E% ^#^	6	2	5	7
Micronutrients					
Cholesterol, mg	-	283	206	266	348
Vitamin A, µg	800 µg ^††^	1436	943	1245	1739
Vitamin C, mg	85 mg ^††^	135	86	124	167
Vitamin D, µg	10 µg ^††^	7	4	6	9
Riboflavin, mg	1.6 mg ^††^	2	2	2	3
Calcium, mg	900 mg ^††^	1151	812	1010	1431
Iron, mg	15 mg ^‡‡^	13	9	12	14
Magnesium, mg	280 mg ^††^	397	301	367	475
Folate, µg	400 µg ^§§^	399	301	397	477

***** Nordic Nutrition Recommendations (NNR) 2012—Integrating Nutrition and Physical Activity. Nordic Council of Ministers: Copenhagen, Denmark, 2014, 5th ed. pp. 19–672. ^‡^ The “dairy products” variable mainly includes milk and yoghurt. ^§^ The “beverages” variable includes water, coffee, tea, mineral water, and alcoholic beverages. ^¶^ The “various intakes” variable includes salty snacks, potato chips, nuts, and various other products that do not fit into the categories above. ^#^ NNR (2012) reference applies to the healthy general adult population, including pregnant women. ** NNR (2012), reference applies to pregnant women. ^††^ NNR (2012), reference applies to pregnant and breastfeeding women. ^‡‡^ NNR (2012), reference applies to women in age groups 18–30 and 31–60 years. ^§§^ NNR (2012), reference applies to women of reproductive age. Number of missing varies between 2 and 5.

## Data Availability

The data presented in this study are available on request from the corresponding author. The data are not publicly available as the participants did not consent to public sharing of data.

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
