# Peer review of "Dietary Intake in Early Pregnancy and Glycemia in Late Pregnancy among Women with Obesity"

_nutrients, 2021, doi:10.3390/nu14010105_

Round 1

Reviewer 1 Report

The manuscript by Garnæs et al. is indeed interesting, relevant and well written.

I however have a couple of points that should be adressed:

  1. The data are based on two RCT's on exercise in pregnancy. Were the women from the current study from the control groups in the TRIP and ETIP studies? This does not seem to be addressed, but is relevant because if the women in the current study are both from the intervention and the control groups it would be difficult to conclude how the dietary intake in the beginning of pregnancy will affect glucose metabolism late in pregnancy. Please elaborate.
  2. Another limitation seems to be the fact that there is no dietary information from late pregnancy, which undoubtedly will contribute to an effect on glucose values. This should also be adressed.

Author Response

Dear Reviewer 1

Please find attached a point-to-point response to your comments. 

Sincerely

Trine Moholdt

Reviewer 2 Report

Since this paper is a study derived from another study, I felt that the current results and conclusions would have little impact on the reader.

My questions are as follows.

  1. Is there any consideration of the impact of pregnancy malady on the results of the dietary studies (especially for early pregnancy)?
  2. I think you need to discuss the fact that there was no significant association between dietary variables and GDM diagnosis on the basis of reference 41.
  3. Wouldn't it be better to include reference 41 in the introduction?
  4. Why are there so many negatives in the Estimated effect of the results in Table 2? There is a report (PMID: 1454084 DOI: 10.1056/NEJM199212313272701) that obese people are more likely to underreport in dietary surveys. Does a large underreporting of subjects affect the results?
  5. I thought it would be better to separate the results of “LatePregnancy” in Table 1 by the presence or absence of GDM.
  6. What is considered to be the cause of the 16 patients who developed GDM?

Author Response

Dear Reviewer 2

Please find our responses to your comments in the attached document.

Sincerely

Trine Moholdt

Round 2

Reviewer 1 Report

The points of criticism have been addressed satisfactory.

Author Response

Thanks

Reviewer 2 Report

I think it's fixed and better.

Author Response

Thanks